# The assessment of the impact of glistening on visual performance in relation to tear film quality

**Amanda Argay**[ORCID]**, Peter Vamosi***

Department of Ophthalmology, Péterfy Sándor Hospital and Traumatology Center, Budapest, Hungary

* vamosipeter@freemail.hu

**Data Availability Statement:** All relevant data are within the paper and its Supporting Information files.

**Funding:** The author(s) received no specific funding for this work.

## Abstract

### Background

The aim of our case control study was to evaluate the impact of glistening and tear film quality on visual performance after implantation of two different hydrophobic acrylic intraocular lenses (IOLs).

### Materials and methods

In our retrospective study we included cataract patients operated between January 1, 2011 and December 31, 2012, with follow-up controls between January 2016 and December 2019. Z-Flex 860FAB (Medicontur) and AcrySof IQ SN60WF (Alcon) monofocal IOLs were implanted during standard phacoemulsification. Best corrected distance visual acuity (BCDVA) and contrast sensitivity were monitored over the post-operative period of up to 6 years. Glistening was evaluated semi-quantitatively with slit-lamp biomicroscopy and quantitatively using Pentacam HR (Oculus). Using HD Analyzer OQAS (Visiometrics), total intra-ocular light diffusion was interpreted with the objective scatter index (OSI) and tear film quality was evaluated with the tear film related objective scatter index (TF-OSI).

### Results

26 eyes implanted with the Z-Flex and 25 eyes with the AcrySof IQ IOLs were included in the analysis. The slit-lamp evaluation of patients with the Z-Flex IOL (0.57 ± 0.60) revealed significantly less glistening (p<0.0001), compared to the AcrySof IQ group (1.82 ± 0.90), and these observations were confirmed by the Pentacam HR analyses, as well (Z-Flex group: 35.1 ± 1.63, Acrysof IQ: 39.6 ± 3.69, p<0.0001). TF-OSI differed between the two sets of patients remarkably (1.53 ± 1.03 vs. 2.51 ± 1.76 for AcrySof IQ and Z-Flex groups, respectively, p = 0.043). Both groups of patients provided similar results of BCDVA and contrast sensitivity.

### Conclusion

Glistening and tear film quality both contribute to visual performance outcomes after cataract surgery. In our study the advantage of less glistening in the Z-Flex IOL might have been

**Competing interests:** The authors have declared that no competing interests exist.

masked by the adverse effects of the more pronounced tear film insufficiency of these patients, compared to the AcrySof IQ group. Among other factors, tear film quality should also be taken into consideration when comparing the impact of glistening on visual quality of patients implanted with different IOLs.

## Background

Glistening describes the phenomenon of light scatter that occurs within intraocular lens (IOL) material in the eyes. An increase in glistening density results in increased light scattering [1,2] and deterioration in the optical quality of the IOL [3]. Glistening is characterized by fluid-filled microvacuoles ranging from 1 to 20 μm in diameter that develop within the material of IOLs [4,5]. It is reported to be present in several IOL materials, although with varying density and degrees [4–8]. Glistening theory suggests that the IOL polymers absorb water when implanted into the wet medium of the eye which leads to phase separation in the IOL material, leaving microvacuoles containing water behind [5,8]. Evaluation of the severity of glistening can be performed by using a subjective semi-quantitative rating scale thereafter [6,7,9–11]. The quantitative measurement of glistening is possible in slit-lamp photographs or by Scheimpflug image analyses through the use of dedicated software such as the EAS-1000 (Nidek Co., Japan) and ImageJ (National Institutes of Health, Bethesda, MD, USA) [10–15]. Although several reports have been published on the phenomenon of glistening, it is still unclear whether glistening has a remarkable impact on visual function and the quality of vision [7,9–31].

Excess intraocular light diffusion causes significant glare which can be quantified by the objective scatter index (OSI), measured by "double-pass" aberrometry [32]. The greater the level of intraocular light scattering, the higher the level of OSI. Light scatter can be caused by tear film instability, lens opacities, microvacuoles or material defects of an implanted IOL, posterior capsule opacification (PCO) and by vitreous floaters [33]. The quality of the tear film has a prominently important role in the post-operative outcome after phacoemulsification with IOL implantation. Post-operative use of artificial tears [34,35], diquafosol ophthalmic solution 3% [36,37] and bandage contact lens [38] was found to improve the quality of the tear film in pseudophakic patients. Dry eye was reported to be the second or third most important reason of dissatisfaction after multifocal intraocular lens implantation [39,40]. The tear film-related objective scatter index (TF-OSI) is a quantitative and objective measure of tear-film related vision quality. The optical quality loss due to the degradation of the tear film can be evaluated also with the double-pass technique [41]. An increased TF-OSI score may contribute to image aberration as a result of impaired tear film break-up [41,42].

To our knowledge, this is the first report in the literature to discuss the visual performance of patients with different degrees of glistening and tear film conditions. The aim of our present study was to compare the degree of glistening and to assess its impact on visual quality in two different one-piece hydrophobic acrylic IOLs, using both a subjective and an objective examination method, and also to evaluate the tear film quality at the same time.

## Materials and methods

### Subjects

In this retrospective observational study, the post-operative visual outcomes were evaluated after cataract surgery in 51 pseudophakic eyes from 42 patients. After retrieval, the data were anonymized for the analyses. The study was performed in accordance with the Tenets of the

Declaration of Helsinki [43]. Due to its retrospective nature no approval of the local ethics committee was necessary. At the time of surgery, all patients gave informed consent following the clinical assessment by the surgeon.

In our retrospective chart review we included patients operated between January 1, 2011 and December 31, 2012, with follow-up controls between January 2016 and December 2019. Exclusion criteria were as follows: any severe ocular surface disease, glaucoma, uveitis, clinically significant posterior capsule opacification, corneal or vitreous opacities, previous intraocular surgery except for cataract surgery, any retinal or optic nerve pathology which could influence post-operative visual performance and complicated cataract surgery.

### Surgical procedure and intraocular lenses

Cataract surgeries were performed by the same experienced surgeon (P.V.) with standard phacoemulsification procedure followed by the mono- or binocular in the bag implantation of either the Z-Flex 860FAB (Medicontur Medical Engineering Ltd.; Zsámbék, Hungary), or the AcrySof® IQ SN60FW (Alcon Inc.; Fort Worth, TX, USA) hydrophobic monofocal intraocular lens. (Table 1) The 2.7 mm tunnel wounds were left sutureless in all cases. No intraoperative and/or post-operative complication was registered. The same surgical procedure and pre- and post-operative examination protocols were followed in all cases.

### Tested parameters

Pre-operative uncorrected distant visual acuity (UCDVA), best corrected distant visual acuity (BCDVA) were recorded, slit-lamp biomicroscopy and Goldmann applanation tonometry were performed. Patients included in the study were called for an appointment after an average of 6 years (± 0.5) following the cataract surgery. At the follow-up control UCDVA, BCDVA, contrast sensitivity measurement and slit-lamp biomicroscopy were performed. After complete pupil dilation with 0.5% tropicamide and 10% phenylephrine hydrochloride eyedrops, the presence of glistening was evaluated both semi-quantitatively and quantitatively, the OSI and TF-OSI were assessed as described below. All patients were examined by the same examiner (A.A.).

### Visual acuity and contrast sensitivity examination

Pre-operative UCDVA and BCDVA were evaluated with a standard ETDRS chart. Post-operative UCDVA, BCDVA and mesopic contrast sensitivity measurements were assessed with a CSV-1000 System (VectorVision, Greenville, Ohio, USA) using the ETDRS (Early Treatment

**Table 1. Characteristics of the intraocular lenses used in the study.**

| Characteristic | Z-Flex 860FAB | AcrySof IQ SN60WF |
|---|---|---|
| Optic material | Hydrophobic acrylic copolymer | Hydrophobic acrylate/methacrylate copolymer |
| Refractive index | 1.47 | 1.55 |
| Abbe number | 58 | 37 |
| Optic design | Biconvex, square edge, anterior and posterior aspheric surface | Biconvex, square edge, anterior and posterior aspheric surface |
| Optic diameter (mm) | 6.0 | 6.0 |
| Length (mm) | 13.0 | 13.0 |
| Haptic configuration | Double C-loop | Modified L |
| Haptic angulation (˚) | 0˚ with posterior vaulting | 0˚ |
| Ultraviolet filter | Yes | Yes + blue light filter |
| A-constant (SRK/T) | 119.1 | 119.0 |

of Diabetic Retinopathy Study) chart and the CSV-1000E chart at 2.44 m with spatial frequencies of 3, 6, 12 and 18 cycles/degree (CPD). The mesopic contrast sensitivity measurements were performed both under non-glare and glare conditions.

## The assessment of glistening

Prior to pupil dilation, pupil size was measured under mesopic conditions using the Pentacam HR Scheimpflug camera (Oculus Optikgeräte GmbH; Wetzlar, Germany) in high resolution front iris camera mode. After pupil dilation the glistening was graded semi-quantitatively by slit-lamp biomicroscopy and scored on a scale from 0 to 3 (from no to severe glistening, respectively), as described in previous studies [6,7,9–11].

Quantitative assessment of glistening was performed by Scheimpflug images of the IOL registered with the Pentacam HR device using the "25 pictures" program mode under mesopic conditions [10]. The mean value of scattering, representing the degree of glistening inside and under the surface of the IOL optic was measured. Values were referred to the brightness or intensity of scattered light on a scale from 0 (black) to 255 (white). Data were imported into the ImageJ digital image processing program (National Institutes of Health, Bethesda, MD, USA) for the objective analyses of glistening in a 1.5 mm zone around the visual axis. Light scattering was evaluated at the anterior and posterior IOL surfaces separately (surface light scattering), and also within the IOL. Surface light scattering is believed to be the result of phase separation of water molecules at the IOL surfaces; however, the mechanism is different from the development of glistening, which is usually present within the IOL material [18]. Thus, calculation of the amount of glistening was performed by omitting the surface light scattering.

## Objective scatter index (OSI) analysis

For quantitative analysis of the intraocular light scattering an objective and quantitative double-pass wavefront device, the HD Analyzer OQAS (Visiometrics S.L., Cerdanyola del Vallès, Spain) was used [19,44]. Prior to the measurements, the patient's cylindrical error was corrected with a trial lens. Spherical refractive errors were automatically compensated by the device itself. The OSI assessment is an objective evaluation of the scattering degree caused by the loss of transparency of one or more of the ocular structures. The higher the OSI value, the higher the level of intraocular scattering which in turn results in lower quality of vision.

## Tear film objective scatter analysis

For the quantitative analysis of the tear film the aforementioned HD Analyzer OQAS was used, employing the same methodology as described above. During the tear film analysis sequence, the patient was requested to casually look at the target. The measurement consisted of recording double-pass images every 0.5 seconds with blinks in the 6th and 14th seconds until a 20 seconds capture has been completed. In this way, the system recorded 40 images, showing the optical quality evolution during those 20 seconds. Comparing the tear film quality of the eyes, the TF-OSI value was used, which was automatically generated by the program. The higher the TF-OSI, the greater the level of tear film scattering is, leading to lower image quality in the eye.

## Statistical analysis

All data were analysed by using the GraphPad Prism 7.04 statistical software (GraphPad Software, San Diego, CA, USA). Pre- and post-operative data of 51 eyes of 42 patients were included in the analyses. Descriptive statistics (mean, standard deviation, median, minimum, maximum, 95% confidence interval) were calculated in all cases. All variables were tested for

normal distribution using the D'Agostino & Pearson test. Depending on the results, comparisons between matching pre- and post-operative variables, or between the two study groups were performed using either the unpaired two-tailed t-test (in case of normal distribution) or the Wilcoxon matched-pairs signed rank test (when non-parametric test was required). Frequency distributions of specific data in the two groups were compared by the Kolmogorov-Smirnov test. Non-parametric Spearman correlation analyses were performed to reveal the possible correlation between the degree of glistening and parameters indicating visual quality. The results of different glistening-examination techniques were compared using a multiple t-test with the Holm-Sidak method. All visual acuities presented are expressed in logMAR, and were measured under photopic conditions. P values of 0.05 or less were considered to be statistically significant in all cases.

## Results

### Patient demographics

Altogether 51 eyes were included in the evaluations. The pre-operative demographic characteristics of the two examined groups, implanted with either the Z-Flex 860FAB (n = 26), or the AcrySof IQ SN60WF lens (n = 25) are indicated in Table 2. Significant difference was observed in the mean age of the two study groups (p = 0.019), as the patients in the Z-Flex group were approximately 5 years older. Average axial length, UCDVA and BCDVA were similar. The average power of implanted IOLs was slightly higher in the AcrySof IQ group.

### Post-operative visual outcomes

There were no intraoperative complications or any adverse events during the follow-up period, except for PCO, which was treated by laser capsulotomy before the post-operative examinations. The mesopic pupil size was measured in the X and Y axes by Pentacam HR in both groups of patients. The pupil sizes in the X (Acrysof IQ: 3.17mm, Z-Flex: 3.12mm) and Y axis (Acrysof IQ: 3.22mm, Z-Flex: 3.13mm) were similar in both groups (X axis p = 0.789, Y axis p = 0.645). We did not find any significant difference in the spherical and cylindrical manifest refractions, the spherical equivalent of manifest refraction and in the UCDVA and BCDVA between the two groups (Table 3). The mesopic contrast sensitivities in different spatial frequencies were similar in both groups, both under non-glare and glare conditions (Fig 1).

### The evaluation of glistening

According to the subjective semi-quantitative measurement of glistening, the Z-Flex IOL was shown to contain significantly less microvacuoles, compared to the Acrysof IQ IOL. The mean

**Table 2. Pre-operative data: Demographics, AXL = axial length, UCDVA = uncorrected distance visual acuity, BCDVA = best corrected distance visual acuity, logMAR = logarithm of the minimum angle of resolution, IOL = intraocular lens, D = dioptre.**

| Data | Z-Flex 860FAB | | AcrySof IQ SN60WF | | Significance (p) |
|---|---|---|---|---|---|
| | Mean ± SD | Range | Mean ± SD | Range | |
| Age (years) | 71.9 ± 5.3 | 64–81 | 66.6 ± 8.4 | 50–79 | **0.019** |
| Female | 17 (77.3%) | | 12 (60.0%) | | |
| Male | 5 (22.7%) | | 8 (40.0%) | | |
| AXL (mm) | 23.65 ± 1.07 | 22.39–26.95 | 23.24 ± 0.78 | 21.42–24.62 | 0.338 |
| UCDVA (logMAR) | 0.68 ± 0.37 | 1.7–0.1 | 0.78 ± 0.45 | 1.7–0.3 | 0.546 |
| BCDVA (logMAR) | 0.35 ± 0.36 | 1.7–0.0 | 0.48 ± 0.45 | 1.7–0.0 | 0.218 |
| IOL Power (D) | +20.4 ± 2.69 | +12.0 - +25.0 | +21.7 ± 1.98 | +17.0 - +25.0 | **0.034** |

**Table 3. Post-operative visual outcomes.** m = manifest, D = dioptre, SE = spherical equivalent, UCDVA = uncorrected distance visual acuity, BCDVA = best corrected distance visual acuity, logMAR = logarithm of the minimum angle of resolution.

| Data | Z-Flex 860FAB | | AcrySof IQ SN60WF | | Significance (p) |
|---|---|---|---|---|---|
| | Mean ± SD | Range | Mean ± SD | Range | |
| m. spherical refraction (D) | +0.51 ± 0.50 | -0.25 - +1.5 | +0.49 ± 0.70 | -0.75 - +2.5 | 0.999 |
| m. cylindrical refraction (D) | -0.36 ± 1.12 | -2.0 - +1.5 | -0.63 ± 0.96 | -3.5 - +0.75 | 0.406 |
| SE of manifest refraction (D) | +0.36 ± 0.65 | -1.0 - +1.75 | +0.18 ± 0.70 | -1.5 -+1.5 | 0.356 |
| UCDVA (logMAR) 6 years postop. | 0.19 ± 0.16 | 0.5–0.0 | 0.14 ± 0.17 | 0.6–0.0 | 0.361 |
| BCDVA (logMAR) 6 years postop. | 0.01 ± 0.03 | 0.1–0.0 | 0.02 ± 0.06 | 0.2–0.0 | >0.999 |

severity of glistening based on subjective assessment was 0.57 ± 0.60 in the Z-Flex, and 1.82 ± 0.90 in the Acrysof IQ group, and the difference was highly significant (p<0.0001). The objective measurement of glistening based on Scheimpflug image analysis clearly confirmed the former results, the Z-Flex IOL was characterized by significantly less glistening (35.1 ± 1.63) than the Acrysof IQ (39.6 ± 3.69), (p<0.0001). The results regarding the glistening of the two IOLs were shown on Fig 2A and 2B.

The comparison of the subjective and objective glistening-assessment techniques was performed after pooling the data from both groups (n = 51). We could reveal that the results of both methods are statistically not different (p = 0.561), whereas the results of the two measurement techniques showed a strong correlation with each other (Spearman r = 0.448; p = 0.001; Fig 2C).

## OSI comparison

The quality of vision expressed as OSI was not significantly different in the two sets of patients: compared to a mean value of 2.42 ± 1.69 of the Acrysof IQ group, a similar value of 2.52 ± 1.73 OSI was measured in Z-Flex patients (p = 0.888). A mild correlation was revealed between OSI and BCDVA ($r^2$ = 0.394; p = 0.0625): the worse the OSI was, the lower BCDVA could be measured.

## TF-OSI comparison

The stability of the tear film expressed as TF-OSI differed between the two groups significantly: compared to a mean value of 1.58 ± 1.03 of the Acrysof IQ group, a much higher value of

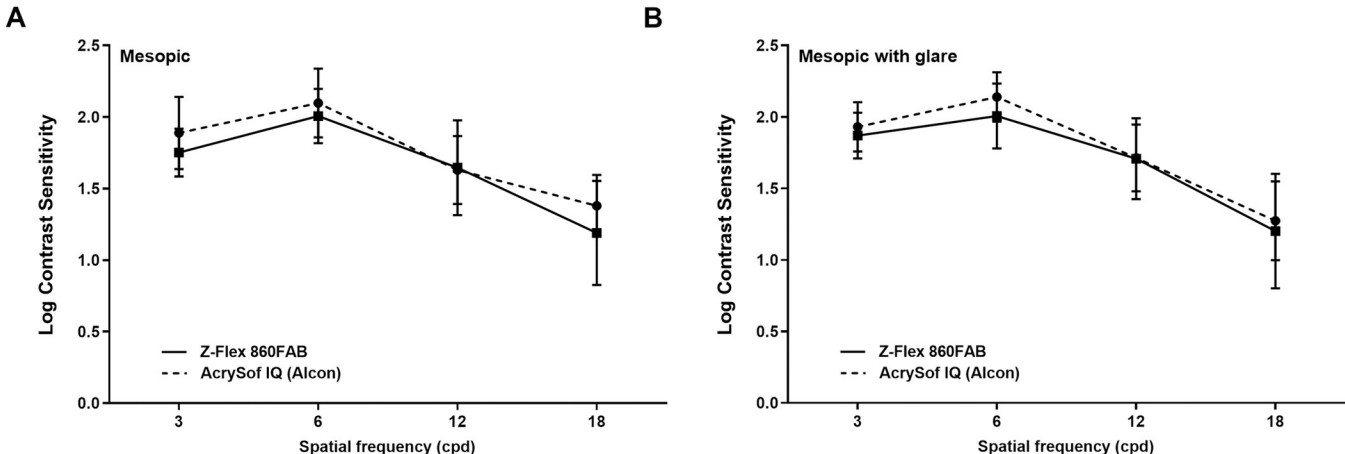

**Fig 1.** The mesopic non-glare (A) and mesopic with glare (B) contrast sensitivity values of two intraocular lenses in different spatial frequencies. There were no statistically significant differences in any spatial frequencies.

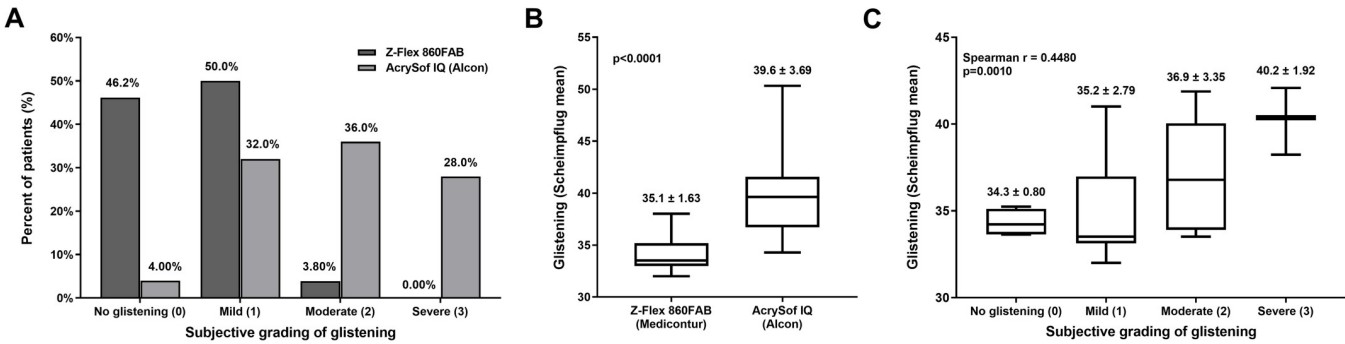

**Fig 2.** (A) Glistening evaluation with the subjective slit-lamp examination method in eyes implanted with the Z-Flex 860FAB or the AcrySof IQ SN60WF IOL. (B) Objective assessment of glistening by Scheimpflug analysis followed by computer-based image analysis in eyes implanted with Z-Flex 860FAB or the AcrySof IQ SN60WF IOL. (C) Correlation analysis revealed a strong correspondence between the results of the two glistening evaluation techniques.

2.79 ± 1.76 was measured in the Z-Flex patients (p = 0.045) (Fig 3A). A significant correlation could be revealed between TF-OSI and BCDVA ($r^2$ = 0.440; p = 0.0354): the worse quality the tear-film had, the lower BCDVA could be measured (Fig 3B).

## The impact of glistening on visual acuity

Because a relatively low number of cases was available in each subjective glistening category (n = 3–11) even after pooling data from all implanted eyes, the possible impact of glistening on visual acuity was assessed by using the results only from the Scheimpflug image analysis. No adverse effect of the presence and severity of glistening could be observed for BCDVA (p = 0.951).

## Discussion

Although several studies have been published on glistening, the impact of this phenomenon on visual function is not completely understood. A considerable number of studies found that

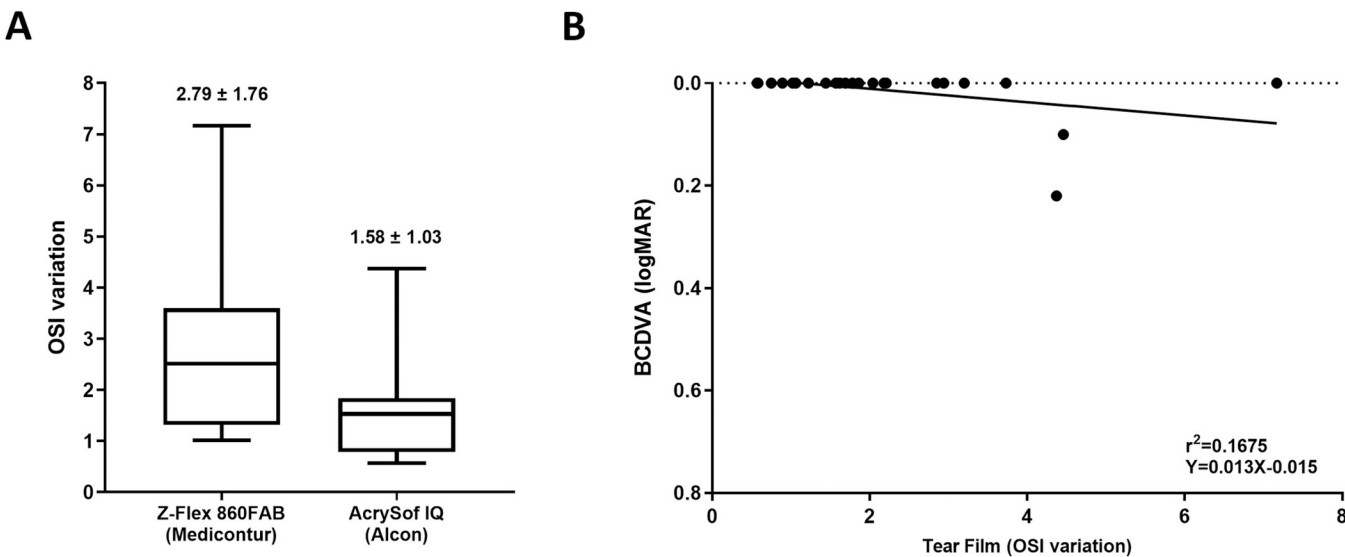

**Fig 3.** (A) The quality of the tear film examined by the HD Analyzer and expressed as TF-OSI values of the different IOLs. (B) Correlation between the TF-OSI and BCDVA.

glistening did not effect either visual acuity [7,9,11,14,15,17–21,23,24,26,31] or contrast sensitivity [11,14,17,18]. However, some authors found diminished visual acuity associated with glistening [16,22,25,29,30]. Other researchers reported decrease of contrast sensitivity in eyes with this phenomenon [7,15,19,23–26,31], especially at high spatial frequencies [19,23–26]. In some cases glistening caused higher loss variance [26] or higher mean deviation [24] in the visual field, or impaired fundus visualization [27]. Examining differences in functional visual acuity, Hiraoka et al. (2017) found that the visual maintenance ratio decreased, while the standard deviation of visual acuity increased in eyes with IOL subsurface nanoglistening [28]. Summarizing the data above, it seems that in most of the recent studies with sophisticated examination methods glistening was associated with some effect on visual function, although it was often only a moderate one. On the other hand, the effect of glistening can be so profound to the visual performance that IOL explantation and replacement with a glistening-free IOL remains the only solution. Previously, we had to explant the IOL from two eyes due to heavily decreased contrast sensitivity and impairment of night driving caused by glistening at a 3+ level on the subjective semi-quantitative rating scale. After IOL exchange and implantation of a new, glistening-free IOL, the patient's complaints completely disappeared (unpublished data). There are a few cases available in the literature reporting about IOL exchange due to clinically significant glistening with similar results [27,29,30].

Glistening can be quantified during slit-lamp examination with results expressed on a semi-quantitative rating scale [6,7,9–11,16]. However, this method can be dependent on the examiner's subjective judgement. For the proper assessment of glistening, the computer analysis of Scheimpflug images seems to be a more objective and examiner-independent technique [10–12,14,15]. In this study we compared the two evaluation methods with each other and found that the results of both methods were statistically identical, in strong correlation with each other. This confirms that even the objective method can deliver reliable data to describe the phenomenon of glistening in a clinical setting. Our results with both techniques showed that the Z-Flex IOL is characterized by significantly less glistening compared to the Acrysof IQ. Our findings are partially in concordance with the results of Behndig and Mönestam who found more glistening in IOLs with a longer post-operative period, and this association was stronger with Scheimpflug quantification than with slit-lamp examination [10]. In another study, the same authors reported a borderline significant association between the subjective grading and total light scattering measured by the Scheimpflug method [11].

The IOL dioptric power of the two examined groups was statistically different. The mean dioptric power of the Acrysof IQ was higher, which may then cause more light scattering because of the higher amount of IOL material that can in turn induce more optical imperfection. This could be in part the reason for the higher glistening in the Acrysof IQ group. These data are in concordance with the findings of some authors [10,11,20,22], while others found no correlation between IOL power and glistening [15,17,18,24].

The tear film is a remarkable factor of post-operative visual performance after phacoemulsification [34–40] which also needs to be considered. Tear film instability and inappropriate quality is known to adversely affect image quality [45,46]. Xue et al. found that vision-related quality of life negatively correlates with dry eye symptoms and positively correlates with visual acuity after phacoemulsification [47]. The tear film has a high impact on optical performance. In our study, patients with any severe ocular surface disease including obvious dry eye disease, were excluded. In spite of that, a significantly different TF-OSI was found between the two sets of patients, a much higher TF-OSI value was measured in Z-Flex patients. A limiting factor of our study is that we focused only for the obvious signs of dry eye and thus the tear break up time (TBUT) and Schirmer tests were not performed and the tear meniscus were not measured. Focusing for the obvious signs might be not enough to assess dry eye disease.

Based on the finding that the Z-Flex IOL had a smaller degree of glistening, it could be expected that visual acuity and/or contrast sensitivity will be superior compared to the Acrysof IQ group. However, all investigated visual outcome parameters were similar in the two examined groups and no statistical difference could be revealed. We suppose that the advantage of less glistening in the Z-Flex IOL might have been masked by the adverse effects of the more pronounced tear film insufficiency of these patients, compared to the AcrySof IQ group. This hypothesis is supported by the fact that the total intraocular light scatter, interpreted by the OSI score was not significantly different between the groups, whereas light scatter caused by the tear film (TF-OSI) was significantly higher in the Z-Flex group. TF-OSI might have at least as dominant an influence in the Z-Flex group as what is caused by glistening in the AcrySof IQ group.

The major limitation of our study is that we were not able to evaluate separately and quantitatively forward light scattering derived by the glistening and the related quality of vision. Also, the OSI measurements are affected by other factors than glistening in the eye structure, such as the tear film, which could also have influence on our results. Finally, another limitation is the relatively low number of eyes in the two groups which warrants further studies in this area.

According to our findings, the thorough assessment of the tear film is also essential when investigating the impact of glistening on visual quality in a certain type of IOL. Due also to this, further prospective studies are needed to measure the impact of glistening.

We believe there is a certain level of glistening above which the optical imperfection leads to clinical consequences regarding the patient's visual performance. A long-term follow-up of this phenomenon is required to measure and define this particular level. Our methodology employing Scheimpflug image analysis followed by computer-based image analysis seems to be an appropriate and reliable method for the objective quantification of IOL glistening that could be used as an objective measure in such studies.

## Supporting information

**S1 Fig.** The mesopic non-glare (A) and mesopic with glare (B) contrast sensitivity values of two intraocular lenses in different spatial frequencies. There were no statistically significant differences in any spatial frequencies.
(DOCX)

**S2 Fig.** (A) Glistening evaluation with the subjective slit-lamp examination method in eyes implanted with the Z-Flex 860FAB or the AcrySof IQ SN60WF IOL. (B) Objective assessment of glistening by Scheimpflug analysis followed by computer-based image analysis in eyes implanted with Z-Flex 860FAB or the AcrySof IQ SN60WF IOL. (C) Correlation analysis revealed a strong correspondence between the results of the two glistening evaluation techniques.
(DOCX)

**S3 Fig.** (A) The quality of the tear film examined by the HD Analyzer and expressed as TF-OSI values of the different IOLs. (B) Correlation between the TF-OSI and BCDVA (n = 23 eyes). TF-OSI = tear film related objective scatter index, BCDVA = best corrected distant visual acuity.
(DOCX)

**S1 Table. Characteristics of the intraocular lenses used in the study.**
(DOCX)

**S2 Table. Pre-operative datas.** Demographics, AXL = axial length, UCDVA = uncorrected distance visual acuity, BCDVA = best corrected distance visual acuity, logMAR = logarithm of the minimum angle of resolution, IOL = intraocular lens, D = dioptre.
(DOCX)

**S3 Table. Post-operative visual outcomes.** m = manifest, D = dioptre, SE = spherical equivalent, UCDVA = uncorrected distance visual acuity, BCDVA = best corrected distance visual acuity, logMAR = logarithm of the minimum angle of resolution.
(DOCX)

**S1 Data.**
(XLSX)

## Acknowledgments

The Authors wish to thank Dr. Gábor Márk Somfai, PhD (Waid and Triemli City Hospital, Zurich, Switzerland) for his valuable help in preparing the manuscript.

## Author Contributions

**Conceptualization:** Amanda Argay.

**Formal analysis:** Amanda Argay.

**Investigation:** Amanda Argay.

**Project administration:** Amanda Argay.

**Supervision:** Peter Vamosi.

**Visualization:** Amanda Argay.

**Writing – original draft:** Amanda Argay.

**Writing – review & editing:** Peter Vamosi.

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
