## [Decision Letter · Decision Letter 0]

22 Jun 2020

PONE-D-20-06388

The assessment of the impact of glistening on visual performance in relation to tear film quality

PLOS ONE

Dear Dr. Argay,

Thank you for submitting your manuscript to PLOS ONE. After careful consideration, we feel that it has merit but does not fully meet PLOS ONE’s publication criteria as it currently stands. Therefore, we invite you to submit a revised version of the manuscript that addresses the points raised during the review process.

Both reviewers have recommended minor changes, which can be done by incorporating changes in the manuscript during this revision. 

We look forward to receiving your revised manuscript.

Kind regards,

Sanjoy Bhattacharya

Academic Editor

PLOS ONE

Journal Requirements:

Reviewers' comments:

Reviewer's Responses to Questions

**Comments to the Author**

1. Is the manuscript technically sound, and do the data support the conclusions?

Reviewer #1: Yes

Reviewer #2: Yes

2. Has the statistical analysis been performed appropriately and rigorously? 

Reviewer #1: Yes

Reviewer #2: Yes

3. Have the authors made all data underlying the findings in their manuscript fully available?

Reviewer #1: Yes

Reviewer #2: Yes

4. Is the manuscript presented in an intelligible fashion and written in standard English?

Reviewer #1: Yes

Reviewer #2: Yes

5. Review Comments to the Author

Reviewer #1: Although in general the manuscript is presented in an intelligible fashion, there are some errors in the use and structure of the English language which make it hard to follow at some points. I advice the authors to improve the flow and readability of the text. Proofreading and better use of the english language should be highly considered. Some of them are highlighted in the comments:

Page 4, Line 18: "clinically PCO", I suggest writing the full name of the disease.

Page 4, Line 20: "influence final visual performance", I don't see "Final" necessary, you may consider adding "The" before it.

Page 5, Line 18: "was recorded", should be: were recorded.

Page 5, Line 22: "was performed", Should be: were performed.

Page 5, Line 22,23: "tropicamide and phenylephrine hydrochloride drops" ,did not mentioned the dose/concentration?

Page 6, Line 6,7: "also under non-glare and glare conditions" , I suggest adding the word "both" to the sentence.

Page 6, Line 15: "25 pictures” instead of „25 pictures”

Page 6, Line 26: "by omitting surface light scattering", I suggest using "The" after the word "omitting".

Page 7, Line 22: "95% confidence intervals", I suggest the word "interval".

Page 8, Line 10: "Fifty one eyes", I suggest sticking to one format: Whether writing numbers in words or numerals rather than randomly mixing the two formats.

Page 8, Line 17: "Data" instead of "Datas".

Page 11, Line 21,22: "authors "I suggest using alternative expression, "researchers" for example or "studies".

Page 12, Line 12: "slit-lamp or slit lamp" instead of "slitlamp".

Fig 2: A,B,C, : looks blurry.

Reviewer #2: This is a very informative paper on the influence of glistening on overall optical quality after post-surgical cataract intervention.

There are minor grammar suggestions for consideration in the attached document.

6. PLOS authors have the option to publish the peer review history of their article (what does this mean?). If published, this will include your full peer review and any attached files.

Reviewer #1: Yes: NAYEF K ALSHAMMARI

Reviewer #2: No

---

## [Author Response · Author response to Decision Letter 0]

1 Sep 2020

Sanjoy Bhattacharya

Academic Editor

PLOS One

Budapest, August 11 2020.

Dear Sanjoy Bhattacharya, 

We would like to thank both reviewers for their valuable and constructive comments on our manuscript entitled "The assessment of the impact of glistening on visual performance in relation to tear film quality”. 

Below you may find our rebuttal letter. We made all modifications and additions suggested by the Reviewers and are convinced that the suggested minor changes helped us to improve the flow and readability of the text. The modifications are highlighted with yellow color in the ‘Revised Manuscript with Track Changes’ file. We also uploaded a clean version of the manuscript. 

We hope that our revised manuscript will meet the high demands of PLOS One.

Sincerely yours,

 Amanda Argay, MD

Peterfy Hospital and Traumatology Center

Department of Ophthalmology

8-20. Péterfy Sándor str.

1076 Budapest, Hungary

Email: argay.amanda@gmail.com

In the following, we answer in detail to the specific comments and additional requests of the Reviewers:

Reviewer #1: Although in general the manuscript is presented in an intelligible fashion, there are some errors in the use and structure of the English language which make it hard to follow at some points. I advice the authors to improve the flow and readability of the text. Proofreading and better use of the english language should be highly considered. Some of them are highlighted in the comments:

We are very grateful to the Reviewer for his detailed correction of grammatical errors. We had the text proofread and corrected in order to improve its readability. 

Page 4, Line 18: "clinically PCO", I suggest writing the full name of the disease.

Correction made.

Page 4, Line 20: "influence final visual performance", I don't see "Final" necessary, you may consider adding "The" before it.

We agree, correction made.

Page 5, Line 18: "was recorded", should be: were recorded.

Correction made.

Page 5, Line 22: "was performed", Should be: were performed.

Correction made.

Page 5, Line 22,23: "tropicamide and phenylephrine hydrochloride drops" ,did not mentioned the dose/concentration?

Correction made, we added the concentration data to the text.

Page 6, Line 6,7: "also under non-glare and glare conditions" , I suggest adding the word "both" to the sentence.

We agree, correction made.

Page 6, Line 15: "25 pictures” instead of „25 pictures”

Correction made.

Page 6, Line 26: "by omitting surface light scattering", I suggest using "The" after the word "omitting".

We agree, correction made.

Page 7, Line 22: "95% confidence intervals", I suggest the word "interval".

We agree, correction made.

Page 8, Line 10: "Fifty one eyes", I suggest sticking to one format: Whether writing numbers in words or numerals rather than randomly mixing the two formats.

We agree, and corrected the number formats.

Page 8, Line 17: "Data" instead of "Datas".

Correction made.

Page 11, Line 21,22: "authors "I suggest using alternative expression, "researchers" for example or "studies".

We agree, correction made.

Page 12, Line 12: "slit-lamp or slit lamp" instead of "slitlamp".

Correction made.

Fig 2: A,B,C, : looks blurry.

We thank the Reviewer for the observation. We resized the figure.

Reviewer #2: This is a very informative paper on the influence of glistening on overall optical quality after post-surgical cataract intervention.

We would like to thank the Reviewer for his positive comment.

There are minor grammar suggestions for consideration in the attached document:

Page 4, Section: Data Availability

Edit: “Yes - all data are fully available without restriction.” *add punctuation

Correction made.

Page 8, Background: “… after implantation of two different hydrophobic acrylic intraocular lenses (IOLs) from 200X until 2019.”

Correction made.

Page 8, “Materials and Methods” *capitalize “Methods” section

Correction made.

Page 8, Materials and Methods

Edit: “Best corrected distance visual acuity (BCDVA) and contrast sensitivity were monitored over a 6-year post-operative period. Glistening was evaluated semi-quantitatively with slit-lamp biomicroscopy and quantitatively using Pentacam HR (Oculus). *break up into two sentences

Correction made.

Page 8, Materials and Methods

Edit: “Both groups of patients provided similar results of BCDVA and contrast sensitivity.

Correction made.

Page 8, Conclusion

Edit: “Glistening and tear film quality both contribute to visual performance outcomes after cataract surgery.”

Correction made.

Page 9, Background

Edit: “Glistening describes the phenomena of light scatter that occurs within intraocular lens (IOL) material in the eyes. An increase in glistening density results in increased light scattering [1,2] and deterioration in the optical quality of IOL [3].” *changed wording

Correction made.

Page 9, Background

Edit: Glistening theory suggests that the IOL polymers absorb water when implanted into the wet medium of the eye, which leads to phase separation in the IOL material, leaving microvacuoles containing water behind [5,8].

Correction made.

Page 9, Background

Edit: The quantitative measurement of glistening is possible in slit-lamp

photographs or by Scheimpflug image analyses through the use of dedicated software such as X, Y, Z [10-15].

Correction made.

Page 9, Background

Edit: Although several reports have been published on the phenomenon of glistening, it is still unclear whether glistening has a remarkable impact on visual function and the quality of vision [7,9-31]. 

Correction made.

Page 9, Background

Edit: Light scatter can be caused by tear film instability, lens opacities, microvacuoles or material defects of an implanted IOL, posterior capsule opacification (PCO), and by vitreous floaters [33].

Correction made.

Page 9, Background

Edit: An increased TF-OSI score may contribute to image aberration as a result of impaired tear film break-up [41,42].

Correction made.

Page 9, Background

Edit: To our knowledge, this is the first report in the literature to discuss the visual performance of patients with different degrees of glistening and tear film conditions.

Correction made.

Page 10, Materials and Methods>Subjects

Edit: In this retrospective observational study, the cataract surgery post-operative visual outcomes of 51 pseudophakic eyes from 42 patients were evaluated.

Correction made.

Page 11, Tested parameters

Edit: Pre-operative uncorrected distance visual acuity (UCDVA), best corrected distance visual acuity (BCDVA) were recorded, slit-lamp biomicroscopy and Goldmann applanation tonometry were performed. … At the follow-up control UCDVA, BCDVA, contrast sensitivity measurement and slit-lamp biomicroscopy were performed.

Correction made.

Page 12, Tested parameters

Edit: Pre-operative UCDVA and BCDVA were evaluated with a standard ETDRS chart. Post-operative UCDVA, BCDVA and mesopic contrast sensitivity measurements were assessed with a CSV-1000 System (VectorVision, Greenville, Ohio, USA) 

Correction made.

Page 12, The assessment of glistening

Edit: Quantitative assessment of glistening was performed by Scheimpflug images of the IOL registered with the Pentacam HR device using the “25 pictures” program mode

under mesopic conditions [10].

Correction made.

Page 14, Patient demographics

Edit: Fifty-one eyes were included in the evaluations.

We solved the problem by slightly modifiying the text to „Altogether 51 eyes...” This way we could maintain the uniform layout of the manuscript. 

Page 14, Table 2

Edit: “Pre-operative data”

Correction made.

Page 19, Post-operative (hyphenate), diopter instead of diopter

Correction made.

Page 17, Instead of starting sentence with “because” replace with “Due to” 

Correction made.

Page 17, when citing Hiraoka et al, please include date

Correction made.

Again, we thank the Reviewer for these grammar suggestions. We made all the suggested changes to the manuscript.

Amanda Argay, MD

---

## [Decision Letter · Decision Letter 1]

28 Sep 2020

The assessment of the impact of glistening on visual performance in relation to tear film quality

PONE-D-20-06388R1

Dear Dr. Argay,

We’re pleased to inform you that your manuscript has been judged scientifically suitable for publication and will be formally accepted for publication once it meets all outstanding technical requirements.

Kind regards,

Sanjoy Bhattacharya

Academic Editor

PLOS ONE

Additional Editor Comments (optional):

Reviewers' comments:

Reviewer's Responses to Questions

**Comments to the Author**

1. If the authors have adequately addressed your comments raised in a previous round of review and you feel that this manuscript is now acceptable for publication, you may indicate that here to bypass the “Comments to the Author” section, enter your conflict of interest statement in the “Confidential to Editor” section, and submit your "Accept" recommendation.

Reviewer #1: All comments have been addressed

2. Is the manuscript technically sound, and do the data support the conclusions?

Reviewer #1: Yes

3. Has the statistical analysis been performed appropriately and rigorously? 

Reviewer #1: Yes

4. Have the authors made all data underlying the findings in their manuscript fully available?

Reviewer #1: Yes

5. Is the manuscript presented in an intelligible fashion and written in standard English?

Reviewer #1: Yes

6. Review Comments to the Author

Reviewer #1: In this research article which is named:"The assessment of the impact of glistening on visual performance in relation to tear

film quality", I found the data are supportive, Statistical Analysis performed appropriately, All datas underlying the findings are available, The manuscript is written in standard English and all the corrections has been made.

7. PLOS authors have the option to publish the peer review history of their article (what does this mean?). If published, this will include your full peer review and any attached files.

Reviewer #1: **Yes: **Nayef K Alshammari

---

## [Editor Report · Acceptance letter]

1 Oct 2020

PONE-D-20-06388R1 

The assessment of the impact of glistening on visual performance in relation to tear film quality 

Dear Dr. Argay:

I'm pleased to inform you that your manuscript has been deemed suitable for publication in PLOS ONE. Congratulations! Your manuscript is now with our production department. 

Kind regards, 

on behalf of

Dr. Sanjoy Bhattacharya 

Academic Editor

PLOS ONE